# Defining Global Thresholds for Serum Ferritin: A Challenging Mission in Establishing the Iron Deficiency Diagnosis in This Era of Striving for Health Equity

**DOI:** 10.3390/diagnostics15030289

**Published:** 2025-01-26

**Authors:** Rodolfo Delfini Cancado, Lauro Augusto Caetano Leite, Manuel Muñoz

**Affiliations:** 1Department of Hematology, Faculdade de Ciências Médicas da Santa Casa de Sao Paulo, Sao Paulo 01224-001, Brazil; lauro-acl@hotmail.com; 2Department of Oncology, Faculdade de Ciências Médicas da Santa Casa de Sao Paulo, Sao Paulo 01224-001, Brazil; 3Hospital Samaritano de Sao Paulo, Sao Paulo 01232-010, Brazil; 4Peri-Operative Transfusion Medicine, School of Medicine, 29010 Malaga, Spain; mmunoz@uma.es

**Keywords:** ferritin, iron deficiency, anemia

## Abstract

Iron deficiency (ID) is a critical public health issue globally and the most prevalent cause of anemia. Iron deficiency anemia (IDA) affects approximately 1.2 billion individuals worldwide, and it is estimated that non-anemic iron deficiency (NAID) is at least twice as common as IDA. Yet, there is still uncertainty about how to diagnose it in clinical practice. The serum ferritin (SF) threshold of <15 ng/mL proposed by the World Health Organization (WHO) has been questioned over the last decade. The current SF thresholds are inappropriately low, and, therefore, a large portion of the population at the most significant risk of ID remain undiagnosed and untreated. The correlation between SF, hepcidin, and the physiological upregulation of iron absorption in healthy adults suggests that SF of <50 ng/mL indicates a more precise threshold for diagnosing ID in adults. Therefore, adopting the SF threshold <50 ng/mL would break up the perpetuation of an inequitable cycle of disadvantage for ID individuals, especially among women.

## 1. Introduction

Iron plays a central role in various essential cellular functions, including oxygen transport and exchange, ATP production, protection from oxidative damage, DNA synthesis and repair, cellular oxygen sensing, regulation of gene expression, and amino acid and lipid metabolism. Iron’s ability to accept and donate electrons is essential for participating in several enzymatic reactions [1,2].

The average amount of iron in the human body is approximately 40 mg/kg of body weight and 50 mg/kg of body weight in females and males, respectively. The body’s iron is distributed among hemoglobin (Hb) within erythroid precursors and mature red blood cells (which accounts for more than two-thirds of the body’s iron), myoglobin in muscles, iron-dependent proteins for cellular metabolism, and storage iron (mainly in the liver, spleen, and bone marrow) [3,4]. A minority of the body’s total iron is found in the circulation, where it is bound to transferrin. Iron stored in muscle is less readily available for mobilization than in macrophages or the liver [3,4]. Each ferritin complex can contain up to 4500 iron atoms. This feature enables the cell to store a substantial quantity of iron, thus avoiding free iron which can trigger harmful reactions. At a steady state, the serum ferritin (SF) concentration reasonably reflects the total body iron stores. In a healthy adult, 1 ng/mL of SF indicates 8–10 mg of storage iron, whereas 200–250 mg is required to raise the Hb concentration by one g/dL [3,4].

ID is one of the leading causes of the global disease burden, disproportionately impacting infants and young children, women of reproductive age, and pregnant and postpartum women, mainly from low- and middle-income countries [1,2,3,4]. According to the WHO, it is estimated that nearly 25% of the world’s population (approximately 1.8 billion people) suffer from some form of anemia, with most of these cases associated with ID [5,6,7].

IDA is merely the tip of the iceberg. It is estimated that non-anemic iron deficiency (NAID) is at least twice as common as IDA. Global public health priorities include treating NAID and IDA. WHO has set a goal of reducing the global prevalence of anemia in women of childbearing age by 50% by 2025, and combating ID is a key component of this goal [7]. However, the issue is complex and cannot be corrected using a single approach. Possible public health strategies include health infrastructure and education campaigns, targeted iron fortification, dietary modification, additional iron supplementation for pregnant women, the control of helminths, and the treatment of the diagnosed individual who is self-evident [1,2,7].

Diagnosing ID is a continuous challenge in clinical practice. An adequate SF threshold to correctly recognize ID is essential to avoid progression to IDA and its preventable consequences by treating these populations adequately [7,8,9].

## 2. Serum Ferritin: A Biomarker of Iron Status That Requires Caution in Clinical Decisions

SF is a noninvasive measure of iron stores and is the most commonly utilized biomarker for evaluating ID [1,2,7,8,9]. It is a quantitative, reproducible, sensitive, relatively cheap, and easy-to-perform method. In children, the mean SF concentration is lower than in adults; in adult men, it is two to three times higher than in premenopausal women, and after menopause, these values are similar for both genders. The universally accepted SF values within the normal range are 15–300 ng/mL and 15–200 ng/mL for healthy men and women, respectively [3,4,7,8,9].

It is worth noting that SF is a marker of acute and chronic inflammatory conditions that can be markedly elevated in persons with average iron storage in many different medical conditions, such as rheumatoid arthritis, systemic lupus erythematosus, chronic kidney disease, hepatitis, excessive alcohol consumption, heart failure, acute infection, intestinal disorders such as coeliac disease, obesity, malignancy, trauma, and in surgical patients [1,2,8,9,10,11,12]. When SF is <15 ng/mL, iron levels are no longer observed in bone marrow macrophages, and most patients are already anemic. SF <15 ng/mL is a reliable indicator of absolute ID, irrespective of the presence or absence of inflammatory conditions [6,7,8,9,10].

The prevalence of ID varies significantly with different definitions. The prevalence of NAID is reported in a cohort of 62 685 US and Canadian women from the Hemochromatosis and Iron Overload Screening (HEIRS) Study. Three definitions of NAID were used and are as follows: (1) combined SF <15 ng/mL and transferrin saturation (TSAT) <10% (HEIRS definition), (2) SF <15 ng/mL (WHO definition), and (3) SF <25 ng/mL (a threshold for iron-deficient erythropoiesis (IDE)). The prevalence of ID significantly increased across definitions in the order of HEIRS (3.12%), WHO (7.43%), and IDE (15.33%). Among 40 381 women aged from 25 to 54 years old, ID was observed in HEIRS (4.46%), WHO (10.57%), and IDE (21.23%). Moreover, the relative ID prevalence among the 62 685 women increased 2.4-fold (95% CI, 2.3–2.5; *p* < 0.001) using the WHO definition and increased 4.9-fold (95% CI, 4.7–5.2; *p* < 0.001) using the IDE definition [13].

Data from the National Health and Nutrition Examination Survey (NHANES) was used to study the prevalence of NAID and IDA in US females [14]. Among 12- to 21-year-old US females (*n* = 3490) between 2003 and 2020, ID (SF <25 ng/mL) affected almost 40% and IDA (Hb <12 g/dL and SF <25 ng/mL) 6%, with variation by the SF or Hb thresholds used. Menstruation was a risk factor for both, but more than one-quarter of premenarchal individuals had NAID.

## 3. Absolute Iron Deficiency and Anemia of Inflammation

The two major forms of ID are absolute ID and anemia of inflammation (AI). These forms can manifest either as isolated or may also coexist [1,2,15,16]. Absolute ID, including NAID and IDA, is associated with several causes, such as increased iron requirements (e.g., infants, preschool children, pregnant women), low iron intake (from malnutrition, or for vegetarians and vegans), impaired iron absorption (from bariatric surgery, *Helicobacter pylori* infection, or coeliac disease), or chronic blood loss (from heavy menses, hookworm infestation, anticoagulants, antiplatelet compounds, or frequent blood donation). As stated above, ID is not a diagnosis per se. For this reason, it is essential to consider the patient’s age, gender, clinical history, and symptoms to identify the underlying cause [1,2,15,16].

AI is associated with chronic immune activation, including infections, autoimmune diseases, heart failure, chronic kidney and pulmonary disease, cancer, obesity, and anemia of the elderly. Increased hepcidin inhibits iron absorption by occluding ferroportin and also acts on macrophages to block the release of iron recycled from senescent erythrocytes into the plasma. Iron sequestration into iron storage sites is by far the most critical pathogenic factor in AI resulting in iron-restricted erythropoiesis [17]. Moreover, in states of increased erythropoiesis, such as during therapy with an erythropoiesis-stimulating agent or after significant blood loss, erythropoiesis may become iron-restricted, compromising the mobilization of iron from storage and the Hb synthesis. The ultimate consequence of these functional disturbances of iron homeostasis is AI [17,18,19,20,21].

## 4. Iron Deficiency Anemia and Non-Anemic Iron Deficiency: An Ongoing Misconception

IDA and NAID are distinct conditions, although they are often used as synonymous. The term ID encompasses low iron stores that fail to meet the body’s iron requirements, irrespective of the presence or absence of anemia. NAID may reduce Hb synthesis, but it is only classified as IDA once Hb levels fall below specific cut-off values, such as 130 g/L in males, 120 g/L in non-pregnant females, and 110 g/L in pregnant women [1,2,8].

The symptoms and signs of NAID, IDA, and AI are similar and often overlooked [1,2,21,22]. Thus, laboratory reference ranges are essential for accurately interpreting test results and clinical decision making.

It is noteworthy that it is uncertain to what extent symptoms of anemia are attributed to hypoxia and decreased tissue oxygen tension, as opposed to NAID or AI, which impair mitochondrial function, cellular metabolism, enzyme activities, and neurotransmitter synthesis [21,22,23].

Clinical observations of the symptomatic improvement in women treated for NAID and interventional studies in patients with congestive heart failure both highlight the critical effects of cellular ID, even without the correction of anemia [21,22,23]. Unlike IDA, anemia-related symptoms in AI are often attributed to the underlying disease, which may explain why AI is commonly not treated [21].

## 5. The Lower Limit of Serum Ferritin for Defining Iron Deficiency: An Issue That Must Be Revised

There is an ongoing debate about the level of SF that defines ID. Based on expert recommendations, the WHO defines ID in persons older than five years in global populations as SF is <15 ng/mL or <70 ng/mL in the presence of infection or inflammation [6].

Some evidence suggests that this threshold could overlook up to 50% of patients with ID, particularly premenopausal women who are most vulnerable to the consequences of ID [8,9,14]. Given the real-world evidence that 30% to 50% of children and women have ID, it is clear that the low limit of normality for SF proposed for diagnosing ID is inappropriate and could explain the under-diagnosis and under-recognition of ID worldwide [8,9,14].

The recommended cut-off values for defining ID are low and vary widely across countries according to gender, age, pregnant and non-pregnant women, and the infectious/inflammatory context across specialties and indications [1,2,8,9,10].

A 2017 meta-analysis by Daru et al. [24] reported that a stainable iron score of zero corresponded to an SF of 15 ng/mL or below, while reduced iron stores (stainable iron score 1+) represented a SF of approximately 70 ng/mL. The present report suggests that changes in hemoglobin and red cell indices that could indicate early ID at a cellular level begin to appear at SF concentrations, which were reported to indicate stainable iron stores that are reduced but not yet absent.

Although many laboratory indexes may be helpful for the ID differential diagnostic workup, such as the mean corpuscular volume (MCV), the reticulocyte count and reticulocyte Hb content, the soluble transferrin receptor and its correlation with SF, and hepcidin measurement, based on the real-life evidence, the determination of iron status by using SF and TSAT is the most commonly deployed strategy used in clinical and public health settings. It is a usual practice in many specialties and guidelines to assume that, regardless of the lower normality value of the ferritin test used, SF <30 ng/mL (irrespective of the TSAT value) or SF <100 ng/mL and TSAT <20% are the hallmarks of absolute ID; while SF >100 ng/mL and TSAT <20% usually indicate AI in the context of infection or inflammation [1,2,15,16,17,18,19,20].

## 6. Non-Anemic Iron Deficiency: A Highly Prevalent Condition Among Pregnant Women

The importance of pre-pregnancy body iron reserves cannot be overstated, as there is a significant physiological increase in the requirement for absorbed iron to expand the red blood cell mass of the woman and ensure an adequate iron supply for the function of the placenta and the growing fetus [25,26]. For a normal pregnancy, the total iron needed is approximately 1000–1200 mg. The woman needs 500 mg of iron at conception, which corresponds to SF concentrations of 70-80 ng/mL, to complete a normal gestation without taking iron supplements or developing ID or IDA [24,25,26].

Milman et al. [27] analyzed data from women of reproductive age from more than 15 European countries, including national surveys and relevant clinical studies. Approximately 40–55% of this population had ID (i.e., an SF concentration ≤30 ng/mL). The prevalence of NAID and IDA were 10–32% and 2–5%, depending on the cutoffs used. About 20–35% of European women in their reproductive ages had enough iron stores (SF concentration >70 ng/mL) to complete a pregnancy without supplementary iron [27].

During pregnancy, the WHO recommended SF thresholds for ID of <15 ng/mL for the first trimester of pregnancy, based on expert opinion. There were no recommendations for the second or third trimester. A very recent paper by Mei et al. [28] analyzed the relationship of SF with two independent indicators of the onset of iron-deficient erythropoiesis (Hb and the soluble transferrin receptor) of 1288 pregnant women between the ages of 15 and 49. It excluded those who were experiencing inflammation or potential liver disease. According to the authors, an SF of <15 ng/mL could underestimate the true prevalence of ID throughout pregnancy. Additionally, one out of ten pregnant women would be recognized as ID using SF thresholds of 25 ng/mL during the first trimester and 20 ng/mL during the second and third trimesters [28].

## 7. Non-Anemic Iron Deficiency: A Prevalent Condition Among Women of Reproductive Age Who Are Blood Donors

Blood donation results in a substantial loss of iron (from 200 to 250 mg) at each bleeding procedure (425 to 475 mL) and subsequent iron mobilization from body stores [29]. Cancado et al. [29] observed that blood donation profoundly influences iron stores and is a significant cause of ID in blood donors, particularly among female donors. Among 300 Brazilian blood donors, the prevalence of ID was 11.0%, of whom 5.5% were male and 31.7% female donors; it was higher in regular blood donors than in first-time blood donors for male blood donors (7.6% vs. 0.0%, *p* < 0.05) and female ones (41.5% vs. 18.5%, *p* < 0.05); and it was higher among male blood donors with three or more donations per year (*p* < 0.05) and among female blood donors with two or more donations per year (*p* < 0.05). These results highlight the high prevalence of ID among blood donors, highlighting the need for a more accurate laboratory trial since Hb alone is not enough to identify and exclude blood donors with NAID [29].

Addo et al. [30] determined SF thresholds for ID in healthy individuals based on data from 286 women, aged from 20 to 49 years old, who were first-time or reactivated donors in the Retrovirus Epidemiology Donor Study-II Donor Iron Status (REDS-RISE). The authors concluded that the physiologically based method for the population studied showed a higher SF threshold for ID of <25 ng/mL, compared to the WHO SF guideline of <15 ng/mL [30].

In 2017, the Dutch National Blood Service Sanquin introduced an SF monitoring policy to prevent the negative consequences of temporary deferral due to low Hb and to protect blood donors from developing anemia. The study included 55 644 donors, 11% deferred, and 40% of whom were due to low Hb. Their SF was measured after the donation. Blood donors with SF levels of 15–30 ng/mL and <15 ng/mL could not donate for 6 months and 12 months, respectively. They found 3.8% of males and 3.0% of females with an SF <15 ng/mL and 3.1% of males and 6.2% of females with an SF of 15–30 ng/mL, respectively [31].

## 8. Non-Anemic Iron Deficiency: A Prevalent Condition Among the Surgical Population

It is worth paying attention to the evidence that NAID is a disease in its own right and warrants attention due to the potential danger it poses for patients (e.g., transfusion requirement and a high rate of morbidity and mortality), especially those who undergo major surgical procedures [31,32,33,34,35]. Therefore, it is recommended that early detection and treatment of pre-operative ID and IDA be undertaken. In a large, multicenter cohort of major surgical procedures (*n* = 3342), the overall prevalence of anemia (Hb <130 g/L) was 36%, with differences according to the type of surgery. Iron status parameters were available for 2884 patients. Pre-operative anemia was detected in 986 patients, of whom 69% were women, 5% had low iron stores (SF 30–100 and TSAT >20%), 62% had absolute ID, and 10% had FID (SF >100 ng/mL and TSAT <20%). For those who were not anemic (*n* = 1898), over two-thirds had absolute ID, FID, or low iron stores [31].

Therefore, as a modifiable risk factor, ID should be checked and corrected whenever possible before any major surgical procedure [32,33,34,35].

## 9. Rethinking and Clearing the Rust of Silence About Iron Deficiency Diagnosis

A recent WHO review highlighted the need for further investigation to establish a higher SF cutoff for ID to detect the majority of patients who are indeed iron deficient [7].

In a landmark study by Guyatt et al. [36], the authors concluded that “the traditional cut-off point dividing normal and abnormal, which in most laboratories is between 12 and 20 ng/mL, is not optimal. The likelihood of ID does not start to drop until values are higher than 40 ng/mL for general populations”.

Laurent et al. [37] analyzed 29 guidelines published over the last decade. They proposed an SF concentration cutoff of 100 ng/mL should be considered in current practices and further clinical trials to define an ID in most conditions. The authors suggested the FID which is defined by SF >100 ng/mL and TSAT <20%, whereas absolute ID is defined by SF <100 ng/dL and TSAT <20%. In a systematic review of 55 studies, an SF threshold of <45 ng/mL had a sensitivity and specificity for ID of 85% and 92%, respectively. Conversely, an SF of <15 ng/mL had a sensitivity and specificity of only 59% and 99%, respectively. The authors concluded that an SF threshold of <45 ng/mL was believed to be the most accurate for diagnosing ID [37].

Based on 7980 adults residing in mainland Portugal, the EMPIRE study detected a prevalence of anemia of 20%, with the highest prevalence among pregnant women. Among the anemic individuals, the proportion of IDA cases varied according to the used SF cut-off, for example, 29.0% (SF <15 ng/mL), 54.8% (<30 ng/mL), 75.4% (<50 ng/mL), and 92.5% (<100 ng/mL). Using an SF threshold of 50 ng/mL, ID with or without anemia affected >50% of the study population, regardless of age or ethnicity, with women being more affected than men (64% vs. 42%, respectively). Again, pregnant women showed the highest ID prevalence (41% for SF <15 ng/mL, 63% for SF <30 ng/mL, and 83% for SF <50 ng/mL) [38].

## 10. Are There Any Physiological or Scientific Reasons Not to Adopt a Serum Ferritin Value Cutoff of 50 ng/mL for Adults?

Galetti et al. [15] published an elegant study analyzing the threshold of SF and hepcidin levels to detect ID based on the upregulation of iron absorption. They performed a pooled analysis of their stable iron isotope studies (*n* = 1058) in healthy women that measured iron absorption from labeled test meals providing physiological amounts of iron. Over the entire range of SF values, hepcidin increased linearly with increasing SF. Iron absorption decreased below a threshold hepcidin level of 3.09 nmol/l, above which iron absorption remained stable. Iron absorption increased below the threshold SF of 51.1 ng/mL, above which iron absorption remained stable. The physiological upregulation of iron absorption leads to a threshold SF of <50 ng/mL, corresponding to a threshold hepcidin of <3 nmol/L, indicating a more precise threshold of ID, namely a physiologic SF cutoff of 50 ng/mL, in young women [15].

In a recent retrospective analysis [39], SF reference intervals were studied in 24,812 working-age adults (55% females) with normal red blood cell parameters, STAT >20%, and normal high-sensitivity C-reactive protein. By using a nonparametric method, changes in parameters consistent with the development of ID began to appear at SF <100 ng/mL. The SF at the inflection points for these changes were approximately 45.1 ng/mL for female and 54.6 ng/mL for males. The value of <50 ng/mL was selected as an ID definition, and it is also consistent with reports that physiologic indicators of ID, such as increased gastrointestinal absorption and decreased serum hepcidin levels, begin to appear at an SF of approximately 50 ng/mL [15].

Those who disagree with raising the SF cutoff have voiced concern that this change will dramatically increase the number of women diagnosed with ID. Based on data published by Mean et al. [39], a 50 ng/mL SF cutoff would define nearly half of all females and approximately 14% of males as ID. Adopting the SF threshold <50 ng/mL would have some health policy implications such as the investigation of the underlying causes of ID and the treatment of NAID to avoid progression to IDA and its preventable consequences by treating these populations correctly. However, this highlights the crux of the issue. Given the overwhelming evidence that numerous individuals are ID and symptomatic, adopting a new SF threshold of <50 ng/mL for diagnosing ID in adults could result in a more precise assessment of the global disease burden.

## 11. Conclusions

ID remains a critical public health issue globally. Diagnosing NAID, ID, and FID is a continuous challenge in clinical practice. An adequate SF threshold to properly recognize NAID is essential to avoid progression to IDA and its preventable consequences by treating these populations adequately. Furthermore, identifying individuals and populations with ID is becoming increasingly important. Although well-designed prospective clinical trials are crucial, scientific evidence suggests that adopting an SF threshold <50 ng/mL would also be an early indicator for NAID based on the normalization of iron absorption. The adoption of the SF threshold <50 ng/mL would break up the perpetuation of an inequitable cycle of disadvantage for ID individuals, especially among women.

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
