# Peer review of "Defining Global Thresholds for Serum Ferritin: A Challenging Mission in Establishing the Iron Deficiency Diagnosis in This Era of Striving for Health Equity"

_diagnostics, 2025, doi:10.3390/diagnostics15030289_

Round 1

Reviewer 1 Report

Comments and Suggestions for Authors

The authors present a very well written manuscript on an important and timely subject. The main conclusions that that we should redefine a threshold cutoff of serum ferritin of 50 ng/ml such that all persons below are iron deficient. The manuscript is reasonably well organized with identification of several vulnerable groups and data that supports changing the definition of ID to <50ng/ml. The main deficit in the manuscript is what would be the recommendation of how to manage all individuals with ferritin <50ng/ml. Specific recommendations are delineated below:

Major

1. It is not clear whether all the categories of vulnerable populations would be approached similarly if the threshold was changed. This should be made clear. For example, in the pre-op population, should there be a goal Hb prior to surgery? Should there be a goal ferritin? Are we saying just anything above ferritin 50? In which case a patient would need to be seen in hematology, treated for a few months with PO iron or with IV iron more quickly and then re-evaluated before surgery is scheduled? If the authors are raising these vulnerable populations to note how important it is to identify iron deficiency in them it would then be useful to have their recommendation for how to proceed. Similarly, a likely different approach would be needed for blood donors and that would be determined by male vs female and frequency of donation and type of donation (RBC vs apheresis RBC vs platelet vs plasma). There is also a large body of work in the US (Barbara Bryant, Alan Mast) regarding ferritin testing and iron supplementation in blood donors which would be additionally instrumental towards this point.

2. The authors note that the reason the current threshold is set so low is possibly to avoid overwhelming the process of evaluating iron deficiency. Similar to the question above, would everyone found to have iron deficiency need a colonoscopy (male and female? all age groups?)? Evaluation for a bleeding diathesis in women with heavy menses without anemia but with iron deficiency? And if so, how many colonoscopies are we talking about? How would such work up decisions be made? Conversely, would treating iron deficiency hide the need to evaluate for a colon cancer (for example)? These are likely some of the questions that were struggles during the previous setting of these thresholds. I would suggest making recommendations for men and women separately in response to finding ferritin <50ng/ml.

3. functional iron deficiency needs more explanation. It is presented on page 2 but defined on page 3. the explanation on page 3 is misleading as absolute ID and FID do not co-exist. Absolute iron deficiency and anemia of chronic inflammation may co-exist but the purpose of this idea is to determine whether there is an expectation that anemia would be at least partially treated with iron supplementation and whether PO or IV iron would be most appropriate.

4. page 3, how frequent at DMT1 mutations that they warrant mention here? Are they more common than TMPRSS6 mutations? Please include both as they both impede iron absorption.

5. page 3, FID paragraph is confusing. It may be useful to distinguish it absolute iron deficiency without relation to anemia. I have not seen FID used outside of anemia causes. In fact, the idea of iron "sequestration" on page 6 means it isn't available for erythropoiesis, leading to anemia, although there is no actual deficiency and thus no expectation that iron supplementation would work. Thus, iron restricted erythropoiesis is a more meaningful term instead of FID and thus not meaningful in the context of iron deficiency WITHOUT anemia. The point I believe the authors are trying to make here is that the requirements for iron in the setting of ESA use are high and if ferritin is >50ng/ml and bleeding occurs or ESA is started, ferritin is likely to decrease BECAUSE of bleeding or ESA use. This is not FID but a thoughtful explanation of how iron is used in erythropoiesis to avoid anemia and should be edited as such. For example, page 4, line 170 defines elevated ferritin with decreased TSAT, in the absence of anemia, would not likely be treated. Are the authors suggesting that TSAT <20 requires therapy when ferritin is >100?

6. do the authors mean on page 6 that FID is the equivalent of "iron sequestration"? This section is very confusing since there is new terminology and no connection with terms used to this point. the surgical patients do not come to hematology attention unless they have severe anemia; it would be more clear to state that this should be evaluated even in the absence of anemia (rather than that it will help treat anemia).

7. I think it would be helpful to include a paragraph on why SF <15 ng/ml was originally selected in a more complete discussion. Otherwise, what is stated on page 4, line 162 becomes a circular argument. In other words, if you increase the threshold for defining ID you are going to have more ID identified so how does that make it a "real world" assessment of the prevalence of ID? This is confusing.

Minor

page 3, "pica" should be defined.

page 3, line 133, "class" should be changed to "classified"

page 4, line 178 "However, it occurs relatively late because of the erythrocyte lifespan." This sentence is not clear; to what does it refer? What is "it"? Please edit.

once ESA is defined, please continue to use ESA rather than erythrocyte stimulating agent, as on page 4, line 182. same for TSAT, SF, ID throughout.

page 4, section 5 needs a concluding sentence of some kind.

page 5 line 241, replace "identify" with "define"

page 5, line 248, add "respectively" after 15-30 ng/ml

Author Response

Comments and Suggestions for Authors

Reviewer 1

The authors present a very well written manuscript on an important and timely subject. The main conclusions that that we should redefine a threshold cutoff of serum ferritin of 50 ng/ml such that all persons below are iron deficient. The manuscript is reasonably well organized with identification of several vulnerable groups and data that supports changing the definition of ID to <50ng/ml. The main deficit in the manuscript is what would be the recommendation of how to manage all individuals with ferritin <50ng/ml. Specific recommendations are delineated below:

I appreciate your consideration. The primary goal of this manuscript is to define ID according to the SF. We have highlighted the importance of investigating the cause of ID, but it was not our goal to detail how to do this in this manuscript.

Major

It is not clear whether all the categories of vulnerable populations would be approached similarly if the threshold was changed. This should be made clear. For example, in the pre-op population, should there be a goal Hb prior to surgery? Should there be a goal ferritin? Are we saying just anything above ferritin 50?

The manuscript has been changed, highlighting that the SF < 50 ng/mL should be used to define an ID in adults, especially for women as a high-risk ID population, and situations where results should be viewed cautiously, such as pregnancy, preoperative setting, and blood donation.

We have highlighted the importance of investigating the cause of ID, but it was not our goal to detail the treatment in this manuscript.

In which case a patient would need to be seen in hematology, treated for a few months with PO iron or with IV iron more quickly and then re-evaluated before surgery is scheduled?

As we mentioned before, the primary goal of this manuscript is to define ID according to the SF. We have highlighted the importance of investigating the cause of ID, but it was not our goal to detail how to do and how to manage patient with ID this in this manuscript.

If the authors are raising these vulnerable populations to note how important it is to identify iron deficiency in them it would then be useful to have their recommendation for how to proceed.

As we mentioned before, the primary goal of this manuscript is to define ID according to the SF. We have highlighted the importance of investigating the cause of ID, but it was not our goal to detail how to do and how to manage patient with ID this in this manuscript.

Similarly, a likely different approach would be needed for blood donors and that would be determined by male vs female and frequency of donation and type of donation (RBC vs apheresis RBC vs platelet vs plasma). There is also a large body of work in the US (Barbara Bryant, Alan Mast) regarding ferritin testing and iron supplementation in blood donors which would be additionally instrumental towards this point.

The investigation and management of ID in blood donors have been revised and discussed more intensively in the last years, partly to guarantee the donor’s health by making oral iron supplementation and/or defining the time between donations according to their SF, particularly in women.

As we mentioned before, the primary goal of this manuscript is to define ID according to the SF. We have highlighted the importance of investigating the cause of ID, but it was not our goal to detail how to do and how to manage blood donors with ID this in this manuscript.

The authors note that the reason the current threshold is set so low is possibly to avoid overwhelming the process of evaluating iron deficiency. Similar to the question above, would everyone found to have iron deficiency need a colonoscopy (male and female? all age groups?)? Evaluation for a bleeding diathesis in women with heavy menses without anemia but with iron deficiency? And if so, how many colonoscopies are we talking about? How would such work up decisions be made? Conversely, would treating iron deficiency hide the need to evaluate for a colon cancer (for example)? These are likely some of the questions that were struggles during the previous setting of these thresholds. I would suggest making recommendations for men and women separately in response to finding ferritin <50ng/ml.

I appreciate your consideration. The main goal of this manuscript is to define ID according to the SF. We have highlighted the importance of the investigation of the cause of ID, but it was not our objective in this manuscript to detail how to do this.

  1. functional iron deficiency needs more explanation. It is presented on page 2 but defined on page 3. Corrected

the explanation on page 3 is misleading as absolute ID and FID do not co-exist. Absolute iron deficiency and anemia of chronic inflammation may co-exist but the purpose of this idea is to determine whether there is an expectation that anemia would be at least partially treated with iron supplementation and whether PO or IV iron would be most appropriate.

I appreciate your consideration. The objective of this manuscript is to clarify the definition of ID. This manuscript doesn`t have the goal of talking about treatment.

  1. page 3, how frequent at DMT1 mutations that they warrant mention here? Are they more common than TMPRSS6 mutations? Please include both as they both impede iron absorption.

Excluded DMT1.

  1. page 3, FID paragraph is confusing. It may be useful to distinguish it absolute iron deficiency without relation to anemia. I have not seen FID used outside of anemia causes. In fact, the idea of iron "sequestration" on page 6 means it isn't available for erythropoiesis, leading to anemia, although there is no actual deficiency and thus no expectation that iron supplementation would work. Thus, iron restricted erythropoiesis is a more meaningful term instead of FID and thus not meaningful in the context of iron deficiency WITHOUT anemia. The point I believe the authors are trying to make here is that the requirements for iron in the setting of ESA use are high and if ferritin is >50ng/ml and bleeding occurs or ESA is started, ferritin is likely to decrease BECAUSE of bleeding or ESA use. This is not FID but a thoughtful explanation of how iron is used in erythropoiesis to avoid anemia and should be edited as such. For example, page 4, line 170 defines elevated ferritin with decreased TSAT, in the absence of anemia, would not likely be treated. Are the authors suggesting that TSAT <20 requires therapy when ferritin is >100?
  2. do the authors mean on page 6 that FID is the equivalent of "iron sequestration"? This section is very confusing since there is new terminology and no connection with terms used to this point. the surgical patients do not come to hematology attention unless they have severe anemia; it would be more clear to state that this should be evaluated even in the absence of anemia (rather than that it will help treat anemia).

Itens 5 and 6 corrected

Absolute iron deficiency and anemia of inflammation

The two major forms of ID are absolute ID and anemia of inflammation (AI). These forms can manifest either as isolated or may also coexist.1,2,15,16 Absolute ID, including NAID and IDA, is associated with several causes, such as increased iron requirements (e.g. infants, preschool children, pregnant women), low iron intake (malnutrition, vegetarians, vegans), impaired iron absorption (bariatric surgery, Helicobacter pylori infection, coeliac disease), chronic blood loss (heavy menses, hookworm infestation, anticoagulants, antiplatelet compounds, frequent blood donation). As stated above, ID is not a diagnosis per se. For this reason, it is essential to consider the patient’s age, gender, clinical history, and symptoms to identify the underlying cause.1,2,15,16

AI is associated with chronic immune activation, including infections, autoimmune diseases, heart failure, chronic kidney and pulmonary disease, cancer, obesity, and anemia of the elderly. Increased hepcidin inhibits iron absorption by occluding ferroportin and also acts on macrophages to block the release of iron recycled from senescent erythrocytes into the plasma. Iron sequestration into iron storage sites is by far the most critical pathogenic factor in AI resulting in iron-restricted erythropoiesis.17 Moreover, in states of increased erythropoiesis, such as during therapy with an erythropoiesis-stimulating agent or after significant blood loss, erythropoiesis may become iron-restricted, compromising the mobilization of iron from storage and the Hb synthesis. The ultimate consequence of these functional disturbances of iron homeostasis is AI.17-21

  1. I think it would be helpful to include a paragraph on why SF <15 ng/ml was originally selected in a more complete discussion. Otherwise, what is stated on page 4, line 162 becomes a circular argument. In other words, if you increase the threshold for defining ID you are going to have more ID identified so how does that make it a "real world" assessment of the prevalence of ID? This is confusing.

The lower limit of serum ferritin for defining iron deficiency: an issue that must be revised

There is an ongoing debate about the level of SF that defines ID. Based on expert recommendations, the WHO defines ID in persons older than 5 years in global populations as SF is < 15 ng/mL or < 70 ng/mL in the presence of infection or inflammation6. The recommended cut-off values for defining ID are low and vary widely across countries according to gender, age, pregnant and non-pregnant women, and the infectious/inflammatory context across specialties and indications.1,2,8-10

Over the last decade, the WHO`s SF threshold < 15 ng/mL for the definition of absolute ID has been questioned.6,8,9,14 Some evidence suggests that this threshold could overlook up to 50% of patients with ID, particularly premenopausal women who are most vulnerable to the consequences of ID.8,9,14  Given the real-world evidence that 30% to 50% of children and women have ID, it is clear that the low limit of normality for SF proposed for diagnosing ID is inappropriate and could explain the under-diagnosis and under-recognition of ID worldwide.8,9,14

Although many laboratory indexes may be helpful for the ID differential diagnostic workup, such as mean corpuscular volume (MCV), reticulocyte count and reticulocyte Hb content, soluble transferrin receptor and its correlation with SF, and hepcidin measurement, based on the real-life evidence, determination of iron status by using SF and TSAT are the most commonly deployed strategy used in clinical and public health settings. it is a usual practice in many specialties and guidelines to assume that, regardless of the lower normality value of the ferritin test used, SF <30 ng/mL (irrespective of the TSAT value) or SF < 100 ng/mL and TSAT <20% are the hallmarks of absolute ID; and SF >100 ng/mL and TSAT <20% usually indicate AI in the context of infection or inflammation.1,2,15-20.

Minor

page 3, "pica" should be defined. Excluded

page 3, line 133, "class" should be changed to "classified"  Done

page 4, line 178 "However, it occurs relatively late because of the erythrocyte lifespan." This sentence is not clear; to what does it refer? What is "it"? Please edit.   Excluded

once ESA is defined, please continue to use ESA rather than erythrocyte stimulating agent, as on page 4, line 182. same for TSAT, SF, ID throughout.  Done

page 4, section 5 needs a concluding sentence of some kind. Done

page 5 line 241, replace "identify" with "define"  Done

page 5, line 248, add "respectively" after 15-30 ng/ml Done

Reviewer 2

Comments and Suggestions for Authors

Thank you for the opportunity to review the manuscript "Defining global thresholds for serum ferritin: a challenging mission in establishing the iron deficiency diagnosis in this era of striving for health equity" submitted by Authors: Rodolfo Delfini Cancado, Lauro Augusto Caetano Leite and Manuel Muñoz 

The topic is an important one to highlight. I have some comments for the authors to consider:

In the introduction it would be helpful to more clearly lay out the purpose of the paper and provide a brief outline so that the reader has an idea of what is to come.

Changed

  1. Line 62 - add the word "iron" before fortification and consider adding screening to this sentence.

Changed

Section 2: Do you have any comments on the variability of different assays and how this can also impact validity of testing? 

Line 76 - do you mean sex instead of gender?

Changed

Section 3: Do you think that a discussion of hepcidin should be included in mention of inflammation and iron regulation?

I appreciate your consideration. Hepcidin is a crucial parameter tightly linked with inflammation and iron homeostasis. However, this parameter is not easily available in clinical practice but only in clinical studies. Measurement of hepcidin is not currently recommended for investigating iron deficiency (2C).

Section 5: What about those with ongoing bleeding? They may require a higher ferritin target. 

Yes, the SF < 50 ng/mL proposal as an ID definition is for everyone.

Section 6: consider reviewing/including this literature as well: Teichman J, Nisenbaum R, Lausman A, Sholzberg M. Suboptimal iron deficiency screening in pregnancy and the impact of socioeconomic status in a high-resource setting. Blood Adv. 2021 Nov 23;5(22):4666-4673. doi: 10.1182/bloodadvances.2021004352. PMID: 34459878; PMCID: PMC8759118.

Article included

Section 7: There is Canadian data on this topic as well: 

Goldman M, Uzicanin S, Osmond L, Yi QL, Scalia V, O'Brien SF. Two-year follow-up of donors in a large national study of ferritin testing. Transfusion. 2018 Dec;58(12):2868-2873. doi: 10.1111/trf.14941. Epub 2018 Sep 27. PMID: 30260480.

Article included

Overall, I think there is opportunity to dig a little deeper into inequities surrounding women's health.

I agree with you. We have emphasized in the manuscript the importance of ID in women, but we don`t have enough space to go deep into inequities surrounding women's health in this manuscript.

Acknowledgement section - this is not a research paper with a study design or data analysis so probably needs to be revised.

Reviewer 2 Report

Comments and Suggestions for Authors

Thank you for the opportunity to review the manuscript "Defining global thresholds for serum ferritin: a challenging mission in establishing the iron deficiency diagnosis in this era of striving for health equity" submitted by Authors: Rodolfo Delfini Cancado, Lauro Augusto Caetano Leite and Manuel Muñoz 

The topic is an important one to highlight. I have some comments for the authors to consider:

1. In the introduction it would be helpful to more clearly lay out the purpose of the paper and provide a brief outline so that the reader has an idea of what is to come.

2. Line 62 - add the word "iron" before fortification and consider adding screening to this sentence.

Section 2: Do you have any comments on the variability of different assays and how this can also impact validity of testing? 

Line 76 - do you mean sex instead of gender?

Section 3: Do you think that a discussion of hepcidin should be included in mention of inflammation and iron regulation?

Section 5: What about those with ongoing bleeding? They may require a higher ferritin target. 

Section 6: consider reviewing/including this literature as well: Teichman J, Nisenbaum R, Lausman A, Sholzberg M. Suboptimal iron deficiency screening in pregnancy and the impact of socioeconomic status in a high-resource setting. Blood Adv. 2021 Nov 23;5(22):4666-4673. doi: 10.1182/bloodadvances.2021004352. PMID: 34459878; PMCID: PMC8759118.

Grace TangAndrea LausmanJameel AbdulrehmanJessica PetrucciRosane NisenbaumLisa K. HicksMichelle Sholzberg; Prevalence of Iron Deficiency and Iron Deficiency Anemia during Pregnancy: A Single Centre Canadian Study. Blood 2019; 134 (Supplement_1): 3389. doi: https://doi.org/10.1182/blood-2019-127602

Section 7: There is Canadian data on this topic as well: 

Goldman M, Uzicanin S, Osmond L, Scalia V, O'Brien SF. A large national study of ferritin testing in Canadian blood donors. Transfusion. 2017 Mar;57(3):564-570. doi: 10.1111/trf.13956. Epub 2016 Dec 9. PMID: 27943371.

Goldman M, Uzicanin S, Osmond L, Yi QL, Scalia V, O'Brien SF. Two-year follow-up of donors in a large national study of ferritin testing. Transfusion. 2018 Dec;58(12):2868-2873. doi: 10.1111/trf.14941. Epub 2018 Sep 27. PMID: 30260480.

Overall, I think there is opportunity to dig a little deeper into inequities surrounding women's health.

Acknowledgement section - this is not a research paper with a study design or data analysis so probably needs to be revised.

Author Response

(The authors gave the same response as above.)

Round 2

Reviewer 1 Report

Comments and Suggestions for Authors

The current manuscript has been further improved.

What remains missing is context. Here are some suggestions for improving the impact of the work:

it is important to add to the discussion on how the current cutoffs were arrived at and in the conclusions about what may be the consequences of increasing the number of ID patients. This is not trivial and it should be raised.

The authors have misunderstood that I was suggesting that they lay out HOW to treat. In fact, what is really important is to understand how the changes they are proposing will impact our fractured healthcare system...

For example, should internists start PO iron in all now new cases of ID who were previously not thought to be ID? Or do these patient need to be seen by Hematology? A discussion about how this change will impact healthcare utilization is thoughtful and would prevent many readers from just disregarding the suggestions therein based on a blind spot regarding feasibility.

Another example is we see MRI with iron overload in many patients who recently received IV iron, leading to anxiety, so a comment about working with our radiology colleagues to add this question to their patient questionnaire and view the results in that context.

These are just tip of the iceberg examples. The main message in the conclusions needs to include ideas pertaining to feasibility and collaboration within healthcare to make this change in a thoughtful and effective way.

Also, conclusion section still refers to FID which I think has been removed from the body of the manuscript.

Author Response

Reviewer 1

The authors present a very well written manuscript on an important and timely subject. The main conclusions that that we should redefine a threshold cutoff of serum ferritin of 50 ng/ml such that all persons below are iron deficient. The manuscript is reasonably well organized with identification of several vulnerable groups and data that supports changing the definition of ID to <50ng/ml. The main deficit in the manuscript is what would be the recommendation of how to manage all individuals with ferritin <50ng/ml. Specific recommendations are delineated below:

I appreciate your consideration. The primary goal of this manuscript is to define ID according to the SF. We have highlighted the importance of investigating the cause of ID, but it was not our goal to detail how to do this in this manuscript.

Major

It is not clear whether all the categories of vulnerable populations would be approached similarly if the threshold was changed. This should be made clear. For example, in the pre-op population, should there be a goal Hb prior to surgery? Should there be a goal ferritin? Are we saying just anything above ferritin 50?

The manuscript has been changed, highlighting that the SF < 50 ng/mL should be used to define an ID in adults, especially for women as a high-risk ID population, and situations where results should be viewed cautiously, such as pregnancy, preoperative setting, and blood donation.

We have highlighted the importance of investigating the cause of ID, but it was not our goal to detail the treatment in this manuscript.

In which case a patient would need to be seen in hematology, treated for a few months with PO iron or with IV iron more quickly and then re-evaluated before surgery is scheduled?

As we mentioned before, the primary goal of this manuscript is to define ID according to the SF. We have highlighted the importance of investigating the cause of ID, but it was not our goal to detail how to do and how to manage patient with ID this in this manuscript.

If the authors are raising these vulnerable populations to note how important it is to identify iron deficiency in them it would then be useful to have their recommendation for how to proceed.

As we mentioned before, the primary goal of this manuscript is to define ID according to the SF. We have highlighted the importance of investigating the cause of ID, but it was not our goal to detail how to do and how to manage patient with ID this in this manuscript.

Similarly, a likely different approach would be needed for blood donors and that would be determined by male vs female and frequency of donation and type of donation (RBC vs apheresis RBC vs platelet vs plasma). There is also a large body of work in the US (Barbara Bryant, Alan Mast) regarding ferritin testing and iron supplementation in blood donors which would be additionally instrumental towards this point.

The investigation and management of ID in blood donors have been revised and discussed more intensively in the last years, partly to guarantee the donor’s health by making oral iron supplementation and/or defining the time between donations according to their SF, particularly in women.

As we mentioned before, the primary goal of this manuscript is to define ID according to the SF. We have highlighted the importance of investigating the cause of ID, but it was not our goal to detail how to do and how to manage blood donors with ID this in this manuscript.

The authors note that the reason the current threshold is set so low is possibly to avoid overwhelming the process of evaluating iron deficiency. Similar to the question above, would everyone found to have iron deficiency need a colonoscopy (male and female? all age groups?)? Evaluation for a bleeding diathesis in women with heavy menses without anemia but with iron deficiency? And if so, how many colonoscopies are we talking about? How would such work up decisions be made? Conversely, would treating iron deficiency hide the need to evaluate for a colon cancer (for example)? These are likely some of the questions that were struggles during the previous setting of these thresholds. I would suggest making recommendations for men and women separately in response to finding ferritin <50ng/ml.

I appreciate your consideration. The main goal of this manuscript is to define ID according to the SF. We have highlighted the importance of the investigation of the cause of ID, but it was not our objective in this manuscript to detail how to do this.

  1. functional iron deficiency needs more explanation. It is presented on page 2 but defined on page 3. Corrected

the explanation on page 3 is misleading as absolute ID and FID do not co-exist. Absolute iron deficiency and anemia of chronic inflammation may co-exist but the purpose of this idea is to determine whether there is an expectation that anemia would be at least partially treated with iron supplementation and whether PO or IV iron would be most appropriate.

I appreciate your consideration. The objective of this manuscript is to clarify the definition of ID. This manuscript doesn`t have the goal of talking about treatment.

  1. page 3, how frequent at DMT1 mutations that they warrant mention here? Are they more common than TMPRSS6 mutations? Please include both as they both impede iron absorption.

Excluded DMT1.

  1. page 3, FID paragraph is confusing. It may be useful to distinguish it absolute iron deficiency without relation to anemia. I have not seen FID used outside of anemia causes. In fact, the idea of iron "sequestration" on page 6 means it isn't available for erythropoiesis, leading to anemia, although there is no actual deficiency and thus no expectation that iron supplementation would work. Thus, iron restricted erythropoiesis is a more meaningful term instead of FID and thus not meaningful in the context of iron deficiency WITHOUT anemia. The point I believe the authors are trying to make here is that the requirements for iron in the setting of ESA use are high and if ferritin is >50ng/ml and bleeding occurs or ESA is started, ferritin is likely to decrease BECAUSE of bleeding or ESA use. This is not FID but a thoughtful explanation of how iron is used in erythropoiesis to avoid anemia and should be edited as such. For example, page 4, line 170 defines elevated ferritin with decreased TSAT, in the absence of anemia, would not likely be treated. Are the authors suggesting that TSAT <20 requires therapy when ferritin is >100?
  2. do the authors mean on page 6 that FID is the equivalent of "iron sequestration"? This section is very confusing since there is new terminology and no connection with terms used to this point. the surgical patients do not come to hematology attention unless they have severe anemia; it would be more clear to state that this should be evaluated even in the absence of anemia (rather than that it will help treat anemia).

Itens 5 and 6 corrected

Absolute iron deficiency and anemia of inflammation

The two major forms of ID are absolute ID and anemia of inflammation (AI). These forms can manifest either as isolated or may also coexist.1,2,15,16 Absolute ID, including NAID and IDA, is associated with several causes, such as increased iron requirements (e.g. infants, preschool children, pregnant women), low iron intake (malnutrition, vegetarians, vegans), impaired iron absorption (bariatric surgery, Helicobacter pylori infection, coeliac disease), chronic blood loss (heavy menses, hookworm infestation, anticoagulants, antiplatelet compounds, frequent blood donation). As stated above, ID is not a diagnosis per se. For this reason, it is essential to consider the patient’s age, gender, clinical history, and symptoms to identify the underlying cause.1,2,15,16

AI is associated with chronic immune activation, including infections, autoimmune diseases, heart failure, chronic kidney and pulmonary disease, cancer, obesity, and anemia of the elderly. Increased hepcidin inhibits iron absorption by occluding ferroportin and also acts on macrophages to block the release of iron recycled from senescent erythrocytes into the plasma. Iron sequestration into iron storage sites is by far the most critical pathogenic factor in AI resulting in iron-restricted erythropoiesis.17 Moreover, in states of increased erythropoiesis, such as during therapy with an erythropoiesis-stimulating agent or after significant blood loss, erythropoiesis may become iron-restricted, compromising the mobilization of iron from storage and the Hb synthesis. The ultimate consequence of these functional disturbances of iron homeostasis is AI.17-21

  1. I think it would be helpful to include a paragraph on why SF <15 ng/ml was originally selected in a more complete discussion. Otherwise, what is stated on page 4, line 162 becomes a circular argument. In other words, if you increase the threshold for defining ID you are going to have more ID identified so how does that make it a "real world" assessment of the prevalence of ID? This is confusing.

The lower limit of serum ferritin for defining iron deficiency: an issue that must be revised

There is an ongoing debate about the level of SF that defines ID. Based on expert recommendations, the WHO defines ID in persons older than 5 years in global populations as SF is < 15 ng/mL or < 70 ng/mL in the presence of infection or inflammation6. The recommended cut-off values for defining ID are low and vary widely across countries according to gender, age, pregnant and non-pregnant women, and the infectious/inflammatory context across specialties and indications.1,2,8-10

Over the last decade, the WHO`s SF threshold < 15 ng/mL for the definition of absolute ID has been questioned.6,8,9,14 Some evidence suggests that this threshold could overlook up to 50% of patients with ID, particularly premenopausal women who are most vulnerable to the consequences of ID.8,9,14  Given the real-world evidence that 30% to 50% of children and women have ID, it is clear that the low limit of normality for SF proposed for diagnosing ID is inappropriate and could explain the under-diagnosis and under-recognition of ID worldwide.8,9,14

Although many laboratory indexes may be helpful for the ID differential diagnostic workup, such as mean corpuscular volume (MCV), reticulocyte count and reticulocyte Hb content, soluble transferrin receptor and its correlation with SF, and hepcidin measurement, based on the real-life evidence, determination of iron status by using SF and TSAT are the most commonly deployed strategy used in clinical and public health settings. it is a usual practice in many specialties and guidelines to assume that, regardless of the lower normality value of the ferritin test used, SF <30 ng/mL (irrespective of the TSAT value) or SF < 100 ng/mL and TSAT <20% are the hallmarks of absolute ID; and SF >100 ng/mL and TSAT <20% usually indicate AI in the context of infection or inflammation.1,2,15-20.

Minor

page 3, "pica" should be defined. Excluded

page 3, line 133, "class" should be changed to "classified"  Done

page 4, line 178 "However, it occurs relatively late because of the erythrocyte lifespan." This sentence is not clear; to what does it refer? What is "it"? Please edit.   Excluded

once ESA is defined, please continue to use ESA rather than erythrocyte stimulating agent, as on page 4, line 182. same for TSAT, SF, ID throughout.  Done

page 4, section 5 needs a concluding sentence of some kind. Done

page 5 line 241, replace "identify" with "define"  Done

page 5, line 248, add "respectively" after 15-30 ng/ml Done

Reviewer 2 Report

Comments and Suggestions for Authors

Thank you for resubmitting with changes. Functional IDA as a term can still be used if authors desire - but I appreciate the addition of anemia of inflammation with hepcidin pathway called out.

TSAT miss-spelled in line 325. 

Acknowledgement section may need further revision to reflect that this is a review article and not a research study: Rodolfo Delfini Cançado and Manuel Muñoz performed the research, designed the study, analyzed the data, and wrote the paper. Lauro Augusto Caetano Leite analyzed the data and wrote the paper.

A statement in the introduction about the purpose of this review article and what it will cover would be helpful.

Author Response

Thank you for the opportunity to review the manuscript "Defining global thresholds for serum ferritin: a challenging mission in establishing the iron deficiency diagnosis in this era of striving for health equity" submitted by Authors: Rodolfo Delfini Cancado, Lauro Augusto Caetano Leite and Manuel Muñoz 

The topic is an important one to highlight. I have some comments for the authors to consider:

In the introduction it would be helpful to more clearly lay out the purpose of the paper and provide a brief outline so that the reader has an idea of what is to come.

Changed

  1. Line 62 - add the word "iron" before fortification and consider adding screening to this sentence.

Changed

Section 2: Do you have any comments on the variability of different assays and how this can also impact validity of testing? 

Line 76 - do you mean sex instead of gender?

Changed

Section 3: Do you think that a discussion of hepcidin should be included in mention of inflammation and iron regulation?

I appreciate your consideration. Hepcidin is a crucial parameter tightly linked with inflammation and iron homeostasis. However, this parameter is not easily available in clinical practice but only in clinical studies. Measurement of hepcidin is not currently recommended for investigating iron deficiency (2C).

Section 5: What about those with ongoing bleeding? They may require a higher ferritin target. 

Yes, the SF < 50 ng/mL proposal as an ID definition is for everyone.

Section 6: consider reviewing/including this literature as well: Teichman J, Nisenbaum R, Lausman A, Sholzberg M. Suboptimal iron deficiency screening in pregnancy and the impact of socioeconomic status in a high-resource setting. Blood Adv. 2021 Nov 23;5(22):4666-4673. doi: 10.1182/bloodadvances.2021004352. PMID: 34459878; PMCID: PMC8759118.

Article included

Section 7: There is Canadian data on this topic as well: 

Goldman M, Uzicanin S, Osmond L, Yi QL, Scalia V, O'Brien SF. Two-year follow-up of donors in a large national study of ferritin testing. Transfusion. 2018 Dec;58(12):2868-2873. doi: 10.1111/trf.14941. Epub 2018 Sep 27. PMID: 30260480.

Article included

Overall, I think there is opportunity to dig a little deeper into inequities surrounding women's health.

I agree with you. We have emphasized in the manuscript the importance of ID in women, but we don`t have enough space to go deep into inequities surrounding women's health in this manuscript.

Acknowledgement section - this is not a research paper with a study design or data analysis so probably needs to be revised.